# Size-Dependent Buckling and Post-Buckling Analysis of the Functionally Graded Thin Plate Al–Cu Material Based on a Modified Couple Stress Theory

**DOI:** 10.3390/nano12193502

**Published:** 2022-10-07

**Authors:** Feixiang Tang, Fang Dong, Yuzheng Guo, Shaonan Shi, Jize Jiang, Sheng Liu

**Affiliations:** 1Key Laboratory of Transients in Hydraulic Machinery, Ministry of Education, School of Power and Mechanical Engineering, Wuhan University, Wuhan 430072, China; 2The Institute of Technological Sciences, Wuhan University, Wuhan 430072, China; 3School of Electrical and Automation, Wuhan University, Wuhan 430072, China

**Keywords:** buckling, post-buckling, scale effect, power-law, functionally graded material, critical buckling load, critical buckling displacement

## Abstract

Size-dependent functionally graded material thin plate buckling and post-buckling problems are considered using the framework of the MCST (Modified Couple Stress Theory). Based on modified couple stress theory and power law, the post-buckling deflection and critical buckling load of simply supported functionally graded material thin plate are derived using Hamilton’s minimum potential energy principle. The analysis compares the simulation results of linear buckling and nonlinear buckling. Innovatively, a power-law distribution with scale effects is considered. The influences of scale effect parameters *l* and power-law index parameters *k* on buckling displacement, load, and strain energy of plates have been investigated. In this article, it is found that the critical buckling displacement, critical buckling load, and buckling strain energy increase with increases in the power-law index parameters *k*. The membrane energy decreases as the power-law index parameter increases. If the upper and lower layers are swapped, the opposite result is obtained. In comparison, the scale effect parameter is more influential than the power-law exponent. The critical buckling displacement in the *x*-direction is not affected by scale effects. The critical buckling load, the membrane energy, and buckling strain energy increase as the scale effect parameter increases. Scale effects increase material stiffness compared with traditional theory, and the power-law index parameters affect FGM properties such as elastic modulus, Poisson’s ratio, density, etc. Both scale effects parameters and power-law index parameters have important effects on the mechanical behavior of materials.

## 1. Introduction

MEMS/NEMS (micro-electro-mechanical systems/nano-electro-mechanical systems) devices play a very significant role in high-precision fields, showing unprecedented application prospects in aerospace, consumer electronics, medical equipment, and other fields [1,2,3]. The main structural units of MEMS/NEMS micro-sensor devices are micro-structures such as micro-beams, micro-plates, and micro-shells. The size of the structure is generally in the micrometer or even nanometer scale. A large number of experiments [4,5,6] have shown that when the structural size of the material reaches the micro-nano level, the mechanical properties of the material are greatly different from those of the macro, and the classical theory will no longer be applicable. Functionally graded material is a new type of composite material composed of two or more materials with one-dimensional or multi-dimensional continuous component distribution. The combination of multiple components can achieve excellent performance [7,8,9] such as higher thermal resistance, higher fracture toughness, better corrosion resistance, etc. The application of functionally graded materials in MEMS/NEMS devices is expected to improve device performance. Compared with traditional composite materials, functionally graded materials have their unique advantages. Due to mutation in the material interface of traditional composite materials, the deformation difference caused by the different thermal expansion coefficients of the two materials in the thermal environment eventually generates thermal stress concentration on the device, which may bring failure risks such as buckling, post-buckling, delamination, and even fracture. Compared with traditional composite materials, functionally graded materials have no obvious interface, and their composition and structure show continuous gradient changes [10]. The continuous material distribution can greatly improve the bond strength [11], reduce the crack driving force [12], reduce the residual stress and thermal stress [13], and reduce the risk of stress concentration [14]. The schematic diagram of the cross-section of the functionally graded material distributed along the thickness direction is shown in Figure 1. Functionally graded materials contain micro-porous defects, and porosity is one of the most important parameters affecting the mechanical behavior of FGM (functionally graded materials). Souhir Zghal et al. [15] concluded that the porosity leads to a decrease in the elastic modulus, so the deflection of the FGMs beams increases. The coefficient and distribution of porosity directly affect the mechanical properties of the FGMs. Gao et al. [16] concluded that increased porosity reduces the critical buckling load, reducing the stiffness of the beam. The lowest critical buckling load is obtained with the porosity in the T-3 distribution (symmetric with weaker top and bottom surfaces). Static and dynamic responses of functionally graded micro/nano-plates are the basis for the development of functionally graded MEMS devices. Jung et al. [17] studied the bending and vibration of a FGM micro-plate on a Pasternak elastic foundation by coupling the first-order shear deformation theory and the modified couple stress theory. The changes in the bending deflection and natural frequency of the FGM micro-plate with the foundation parameters, material gradient parameters, and scale effect parameters under uniform load are analyzed. Eshraghi et al. [18] studied the free vibration response of the FGM micro-rectangular plate based on the higher-order shear deformation theory and the modified couple stress theory and obtained the analytical solution of the natural frequency. Song [19] et al. studied the vibration response of the FGM micro-beam in the medium under photothermal excitation, obtained the resonance frequency in different fluid environments, and determined the effect of the gradient factor on the resonance frequency. Recently, functionally graded (FG) grapheme platelets (GPLs) and reinforced composites (FG-GPLRC) have great application potential in MEMS/NEMS devices due to their superior mechanical properties such as high strength, good thermal conductivity, exceptionally high elastic modulus, and large specific surface area [20]. Yang’s team [21,22,23] was the first to propose a multilayer functionally graded graphene platelets reinforced composite material with a layered variation along the thickness direction, and conducted pioneering work on its mechanical properties. The results show that the mechanical properties of FG-GPLRC are significantly better than those of GPLs. Furthermore, incorporating graphene origami into FGMs can also yield unique mechanical properties such as a negative Poisson’s ratio [24], resulting in versatility and superior structural performance. Yang et al. [24] propose an efficient micro-mechanical model based on a genetic programming (GP) algorithm and molecular dynamics (MD) simulation to predict the mechanical properties of graphene origami metal functionally graded materials. Theoretical results show that the proposed functionally graded metamaterial beam achieves significantly improved bending properties.

Compared with macro-devices, micro-devices represent not only a reduction in size but also a substantial change in mechanical properties. The traditional theory is no longer applicable. The traditional couple stress theory [25,26,27] considers the displacement and rotation of the point particle at the same time, but there are many material parameters considered in the scale effect, making it unsuitable for practical engineering applications. In 2002, Yang et al. [28] proposed a modified couple stress theory that only requires one scale parameter, which reduces the difficulty of solving and also has the potential to guide engineering practice. Chen et al. [29] proposed a new modified couple stress theory for anisotropic materials in 2011. In recent years, Richa Priyanka [30] et al. studied the stability and free vibration characteristics of micro-lamination under varying axial loads based on the MCST (modified couple stress theory). The study found that beams with high buckling loads and high natural frequencies under C-C (Clamped-Clamped) support conditions have more significant scale effects. J. N. Reddy et al. [31] verified the applicability of the modified coupled stress theory for web-core sandwich panels. Li et al. [32] designed a standard experimental method to determine the scale effect parameters of the modified couple stress theory. The scale effect parameters were calculated by measuring the vibration frequency of the micro-cantilever beam. The couple stress theory is an effective method to describe the micro size scale effect, and its dimension is at the micron level. However, when the structural size of the material reaches the nanometer level, the specific surface area also needs to be considered, and the influence of the surface effect cannot be ignored. Non-local theory [33,34], strain gradient theory [35] and surface elasticity theory [36] can be used to describe the scale effect and surface effect of nanostructures. Song [37] et al. considered the refraction and reflection of thermal, elastic, and plasma waves in semiconductor microstructures based on nonlocal theory. The research indicates that wave properties show a significant difference between nonlocal theory and classical theory. Li et al. [38] studied the propagation properties of flexural waves in FGM beams based on the theory of nonlocal strain gradients and showed that the optical phase velocities and acoustical phase velocities increase with decreasing nonlocal or material length scale parameters. Y.S. Li et al. [39] studied the bending properties of piezoelectric nano-plates based on surface effects, concluding that surface effects enhance plate stiffness depending on the thickness of the plate. These theories describing micro-scale structures are derived from continuum mechanics as a complement to classical mechanics to describe the properties of micro-scales.

The post-buckling problem is a geometric nonlinear problem [40], which is caused by large out-of-plane deflection. Von Karman was the first to derive nonlinear equilibrium equations for large deflections of plates. L. L. Ke et al. [41,42] studied the post-buckling behavior and nonlinear vibration characteristics of piezoelectric nano-beams under thermo-electro-mechanical loadings based on the Von Karman nonlinear equation. Based on the Von on Karman plate theory, Chen et al. [43] considered the influence of geometric nonlinearity and damage effects on the post-buckling characteristics of micro-plates. The study found that the larger the scale effect parameter under the same load condition, the less destructive the effect on the post-buckling properties. Amin Ghorbani Shenas et al. [44] analyzed the thermal buckling, post-buckling, and free vibration properties of rotating pre-twisted FGM beams. The effects of angular velocity, torsion angle, scale effect parameter, and power-law index on the post-buckling equilibrium path and linear free vibration behavior of FGM beams were investigated. Li et al. [45] studied the influence of scale effects on post-buckling behavior of FGM nano-beams with Von Karman geometric nonlinearity based on nonlocal strain gradient theory, which found that the values of scaling parameters determine stiffness-softening or stiffness-hardening.

According to the above, there are many studies on the mechanical behavior of FGM materials, and power-law functions [46,47,48] are often used to describe the material properties of FGM. Density, elastic modulus, Poisson’s ratio, etc. are inherent properties of materials. When the size of the material reaches the micro-scale, the scale effects cannot be ignored, and the scale effect parameter is also an inherent property of the micro-scale material based on MCST. Therefore, considering the power-law distribution of scale effects is of great significance. However, there is no mathematical model study on the distribution of the power-law function of the scale effect of FGM. In this paper, the equilibrium method based on MCST combined with the power-law is adopted to obtain the elastic modulus and stiffness of a thin FGM plate. The power-law distribution function of the scale effect parameter is introduced into the equilibrium equation. Scale effect parameters of thin FGM plates vary with the thickness and proportions of materials and are uniquely determined. Coupling the modified couple stress theory into the S. Timoshenko et al. [49,50,51] plate theory, the equilibrium equation of the Al–Cu FGM thin plate is obtained. The critical buckling displacement and critical buckling load of the FGM thin plate are obtained according to Hamilton’s principle [52,53]. The changes in bending energy and membrane energy with power-law index *k* and post-buckling stiffness are analyzed. The stability of MEMS/NEMS in multi-field and multi-scale environments is still a challenge [54,55,56,57,58]. The content of this paper is of great significance for MEMS/NEMS manufacturing and stability analysis.

## 2. Formulations and Theories of FGM with MCST

### 2.1. Power-Law of FGM

Considering that the FGM thin plate is composed of Al and Cu, the material changes continuously from Cu on the lower surface to Al on the upper surface. Power-law functions [46,47,48] are often used to describe the material properties of FGM. The gradient change of material properties can be expressed as a power-law function as [46,47,48]:(1)$$P(z)=(P_{U}-P_{L}){\left(\frac{z}{h}\right)}^{k}+P_{L}$$
where P(z) are the material property parameters of the FGM thin plate at *z*, $P_{U}$ and $P_{L}$ are the material property parameters of the upper and lower surfaces, respectively, *k* is a power-law index representing the volume fraction, and *h* is FGM plate thickness. When *k* = 0, the FGM plate consists entirely of upper surface Al, and when *k* is infinite, the material consists entirely of lower surface Cu.

Therefore, Young’s modulus and Poisson’s ratio of the FGM plate according to Equation (1) can be expressed as follows:(2)$$E(z)=(E_{U}-E_{L}){\left(\frac{z}{h}\right)}^{k}+E_{L}$$
(3)$$\upsilon (z)=(\upsilon_{U}-\upsilon_{L}){\left(\frac{z}{h}\right)}^{k}+\upsilon_{L}$$

Considering the scale effect parameter *l* varies with volume fraction, the power-law function of gradient variation introduced into the scale effect parameter *l* is:(4)$$l(z)=(l_{U}-l_{L}){\left(\frac{z}{h}\right)}^{k}+l_{L}$$

### 2.2. FGM Plate Theory Based on MCST

The constitutive equation of FGM plates can be expressed as follows based on MCST:(5)$$\sigma_{ij}=\lambda \left(z\right)\varepsilon_{kk}\delta_{ij}+2\mu \left(z\right)\varepsilon_{ij}$$
(6)$$\varepsilon_{ij}=\frac{1}{2}(u_{i,j}+u_{j,i})$$
(7)$$\theta_{i}=\frac{1}{2}e_{ijk}u_{k,j}$$
(8)$$\chi_{ij}=\frac{1}{2}(\theta_{i,j}+\theta_{j,i})$$
(9)$$m{\left(z\right)}_{ij}=2l{\left(z\right)}^{2}\mu \left(z\right)\chi_{ij}$$
(10)$$\mu \left(z\right)=G\left(z\right)=\frac{E\left(z\right)}{2\left[1+\upsilon \left(z\right)\right]}$$
(11)$$\lambda \left(z\right)=\frac{E\left(z\right)\upsilon \left(z\right)}{\left[1+\upsilon \left(z\right)\right]\left[1-2\upsilon \left(z\right)\right]}$$
where *σ_ij_*, *ε_ij_*, *χ_ij_*, *m_ij_* are expressed FGM plate stress tensor, strain tensor, symmetric curvature tensor, and couple stress tensor, respectively, *θ_i_* is the FGM plate rotation vector and *e_ijk_* is the permutation symbol, *μ* and *λ* are Lame constants, *E* is the elastic modulus, *υ* is the Poisson ratio based on the power-law of FGM, and *l* is the scale parameter of FGM, which is related to the structure size and power-law index *k*.

Consider an FGM thin plate of length *a*, width *b*, and thickness *h* under the external lateral load *q*. The displacement field *u_i_* based on Kirchhoff thin plate theory is given as:(12)$$u(x,y,z)=-z\frac{\partial w}{\partial x}$$
(13)$$v(x,y,z)=-z\frac{\partial w}{\partial y}$$
(14)w(x,y,z)=w(x,y,t)

The bending moment and shear force according to power-law can be expressed as:(15)$$\begin{array}{l} M_{x}=\int_{h}\sigma_{xx}zdz+\int_{h}m_{xy}dz=\frac{-E\left(z\right)h^{3}}{12(1-\upsilon {\left(z\right)}^{2})}(\frac{\partial^{2}w}{\partial x^{2}}+\upsilon \left(z\right)\frac{\partial^{2}w}{\partial y^{2}})\\ -l{\left(z\right)}^{2}G\left(z\right)h(\frac{\partial^{2}w}{\partial x^{2}}-\frac{\partial^{2}w}{\partial y^{2}}) \end{array}$$
(16)$$\begin{array}{l} M_{y}=\int_{h}\sigma_{xx}zdz-\int_{h}m_{yx}dz=\frac{-E\left(z\right)h^{3}}{12(1-\upsilon {\left(z\right)}^{2})}(\frac{\partial^{2}w}{\partial y^{2}}+\upsilon \left(z\right)\frac{\partial^{2}w}{\partial x^{2}})\\ -l{\left(z\right)}^{2}G\left(z\right)h(\frac{\partial^{2}w}{\partial y^{2}}-\frac{\partial^{2}w}{\partial x^{2}}) \end{array}$$
(17)$$\begin{array}{l} M_{xy}=M_{yx}=\int_{h}\sigma_{xy}zdz-\int_{h}m_{xx}dz=\int_{h}\sigma_{yx}zdz+\int_{h}m_{yy}dz\\ =\frac{-E\left(z\right)h^{3}}{12(1-\upsilon {\left(z\right)}^{2})}\frac{\partial^{2}w}{\partial x\partial y}-2l{\left(z\right)}^{2}G\left(z\right)h\frac{\partial^{2}w}{\partial x\partial y} \end{array}$$
(18)$$Q_{x}=\int_{h}\sigma_{xz}dz$$
(19)$$Q_{y}=\int_{h}\sigma_{yz}dz$$

The equilibrium equation can be expressed as follows:(20)$$\frac{\partial M_{x}}{\partial x}+\frac{\partial M_{yx}}{\partial y}=Q_{x}$$
(21)$$\frac{\partial M_{y}}{\partial y}+\frac{\partial M_{xy}}{\partial x}=Q_{y}$$
(22)$$\frac{\partial Q_{x}}{\partial x}+\frac{\partial Q_{y}}{\partial y}+q=0$$

By using Equations (20)–(22).The FGM thin plate governing equation can be expressed as:(23)$$(D\left(z\right)+l{\left(z\right)}^{2}G\left(z\right)h)\nabla^{4}w=q$$
where
(24)$$D\left(z\right)=\frac{E\left(z\right)h^{3}}{12\left(1-\upsilon {\left(z\right)}^{2}\right)}$$
where D(z) is the FGM plate flexural rigidity. Compared with classical governing equations [49,50,51], the flexural rigidity of FGM thin plate based on power-law and MCST follows as:(25)$$D_{1}\left(z\right)=D\left(z\right)+l{\left(z\right)}^{2}G\left(z\right)h$$

## 3. Post-Buckling of FGM Thin Plate

Consider the geometry and force diagram of the FGM thin plate as shown in Figure 2. The FGM thin plate is of length *a*, width *b*, thickness *h*, and force *P*. The displacement in the *x*-direction under the pressure of *P* is *u_x_*; the displacement in the *y*-direction is *u_y_*, and the deflection is *w*_0_.

The expression of displacement function of the FGM thin plate is as follows:(26)$$u_{x}=u_{0}\left(1-\frac{x}{a}\right)$$
(27)$$u_{y}=\upsilon \left(z\right)u_{0}\frac{y}{a}+f\left(x\right)$$

Assume that the flexural deflection is a double harmonic function as:(28)$$w=w_{0}\sin\frac{\pi x}{a}\sin\frac{\pi y}{b}$$

According to the Von Karman nonlinear equation, the strain of a thin plate is expressed as:(29)$$\varepsilon_{xx}=\frac{\partial u}{\partial x}+\frac{1}{2}{\left(\frac{\partial w}{\partial x}\right)}^{2}=-\frac{u_{0}}{a}+\frac{w_{0}{}^{2}}{2}{\left(\frac{\pi }{a}\right)}^{2}\cos^{2}\frac{\pi x}{a}\sin^{2}\frac{\pi y}{a}$$
(30)$$\varepsilon_{yy}=\frac{\partial u_{y}}{\partial x}+\frac{1}{2}{\left(\frac{\partial w}{\partial y}\right)}^{2}=\nu \frac{u_{0}}{a}+f^{\prime }(x)+\frac{w_{0}{}^{2}}{2}{\left(\frac{\pi }{a}\right)}^{2}\sin^{2}\frac{\pi x}{a}\cos^{2}\frac{\pi y}{a}$$
(31)$$\varepsilon_{xy}=\frac{1}{2}\left(\frac{\partial u_{x}}{\partial y}+\frac{\partial u_{y}}{\partial x}\right)+\frac{1}{2}\frac{\partial w}{\partial x}\frac{\partial w}{\partial y}=f^{\prime }(x)+\frac{w_{0}{}^{2}}{2}{\left(\frac{\pi }{a}\right)}^{2}\cos\frac{\pi x}{a}\sin\frac{\pi y}{a}\sin\frac{\pi x}{a}\cos\frac{\pi y}{a}$$

The curvature components can be defined as follows:(32)$$\kappa_{xx}=-\frac{\partial^{2}w}{\partial x^{2}}=w_{0}{\left(\frac{\pi }{a}\right)}^{2}\sin\frac{\pi x}{a}\sin\frac{\pi y}{b}$$
(33)$$\kappa_{yy}=-\frac{\partial^{2}w}{\partial y^{2}}=w_{0}{\left(\frac{\pi }{a}\right)}^{2}\sin\frac{\pi x}{a}\sin\frac{\pi y}{b}$$
(34)$$\kappa_{xy}=\kappa_{yx}=-\frac{\partial^{2}w}{\partial x\partial y}=-w_{0}{\left(\frac{\pi }{a}\right)}^{2}\cos\frac{\pi x}{a}\cos\frac{\pi y}{b}$$

The total energy at the location $\left(u_{0},w_{0}\right)$ is: (35)$$\prod \left(u_{0},w_{0}\right)=U_{b}+U_{m}-Pu_{0}$$
where $U_{b}$ and $U_{m}$ are strain and membrane energies, respectively. According to references [49,50,51], the FGM thin plate strain energy can be seen as:(36)$$U_{b}=\frac{D_{1}\left(z\right)}{2}\int_{0}^{a}\int_{0}^{a}\left[{\left(\kappa_{xx}+\kappa_{yy}\right)}^{2}-2\left(1-\upsilon \left(z\right)\right)\left(\kappa_{xx}\kappa_{yy}-\kappa_{xy}{}^{2}\right)\right]dxdy$$

By using Equations (32)–(34) in Equation (36), the FGM thin plate strain energy can be derived as:(37)$$U_{b}=\frac{D_{1}\left(z\right)}{2}\int_{0}^{a}\int_{0}^{a}{\left(\kappa_{xx}+\kappa_{yy}\right)}^{2}dxdy=2D_{1}\left(z\right){\int_{0}^{a}\int_{0}^{a}w_{0}{}^{2}\left(\frac{\pi }{a}\right)}^{4}\sin^{2}\frac{\pi x}{a}\sin^{2}\frac{\pi y}{a}dxdy$$

Reducing Equation (37) results in the following:(38)$$U_{b}=2D_{1}\left(z\right)w_{0}{}^{2}{\left(\frac{\pi }{a}\right)}^{4}\int_{0}^{a}\frac{1-\cos\frac{2\pi }{a}x}{2}dx\int_{0}^{a}\frac{1-\cos\frac{2\pi }{a}y}{2}dy=\frac{D_{1}\left(z\right)}{2}\frac{\pi^{4}}{a^{2}}w_{0}{}^{2}$$

More specifically, the strain energy expression considering the MCST of the FGM thin plate is as follows:(39)$$U_{b}=\frac{1}{2}\left\{\frac{\left((E_{U}-E_{L}){\left(\frac{z}{h}\right)}^{k}+E_{L}\right)h^{3}+6h{\left((l_{U}-l_{L}){\left(\frac{z}{h}\right)}^{k}+l_{L}\right)}^{2}\left[1-\left((\upsilon_{U}-\upsilon_{L}){\left(\frac{z}{h}\right)}^{k}+\upsilon_{L}\right)\right]}{12\left[1-{\left((\upsilon_{U}-\upsilon_{L}){\left(\frac{z}{h}\right)}^{k}+\upsilon_{L}\right)}^{2}\right]}\right\}w_{0}{}^{2}\frac{\pi^{4}}{a^{2}}$$

The membrane energy expression is as follows:(40)$$U_{m}=\int_{0}^{a}\int_{0}^{a}\frac{C_{1}\left(z\right)}{2}\left[{\left(\varepsilon_{xx}+\varepsilon_{yy}\right)}^{2}-2\left(1-\upsilon \left(z\right)\right)\left(\varepsilon_{xx}\varepsilon_{yy}-\varepsilon_{xy}{}^{2}\right)\right]dxdy$$
where
(41)$$C_{1}=\frac{E\left(z\right)h}{1-\upsilon {\left(z\right)}^{2}}$$

Substituting Equations (29)–(31) into Equation (40) yields the solution for membrane energy.
(42)$$U_{m}=\frac{C_{1}\left(z\right)}{2}\left[\left(1-\upsilon {\left(z\right)}^{2}\right)u_{0}{}^{2}-\frac{2u_{0}w_{0}^{2}\left(1-\upsilon {\left(z\right)}^{2}\right)}{8a}+\frac{\left(3-2v\right)\pi^{4}w_{0}{}^{4}}{64a^{2}}\right]$$

A more specific expression is shown below:(43)$$\begin{array}{l} U_{m}=\frac{\left((E_{U}-E_{L}){\left(\frac{z}{h}\right)}^{k}+E_{L}\right)+\frac{6{\left((l_{U}-l_{L}){\left(\frac{z}{h}\right)}^{k}+l_{L}\right)}^{2}\left[1-\left((\upsilon_{U}-\upsilon_{L}){\left(\frac{z}{h}\right)}^{k}+\upsilon_{L}\right)\right]}{h^{2}}}{2\left[1-{\left((\upsilon_{U}-\upsilon_{L}){\left(\frac{z}{h}\right)}^{k}+\upsilon_{L}\right)}^{2}\right]}\\ \times \left[\left[1-{\left((\upsilon_{U}-\upsilon_{L}){\left(\frac{z}{h}\right)}^{k}+\upsilon_{L}\right)}^{2}\right]u_{0}{}^{2}-2\left[1-{\left((\upsilon_{U}-\upsilon_{L}){\left(\frac{z}{h}\right)}^{k}+\upsilon_{L}\right)}^{2}\right]\frac{\pi^{2}}{8}\frac{u_{0}}{a}w_{0}{}^{2}+\left(3-2\left((\upsilon_{U}-\upsilon_{L}){\left(\frac{z}{h}\right)}^{k}+\upsilon_{L}\right)\right)\frac{\pi^{4}}{64}\frac{w_{0}{}^{4}}{a^{2}}\right] \end{array}$$

According to Hamilton’s principle [52,53], among all possible displacement fields for deformation, the real displacement field minimizes the total potential energy.
(44)δ∏=0

The equilibrium at the *x*-directional displacement of $u_{0}$ and out-of-plane deflection of $w_{0}$ requires that the first-order energy partial derivatives of different directions are all zero. Thus, we can obtain the following results:(45)$$\frac{\partial \prod }{\partial u_{0}}=0$$
(46)$$\frac{\partial \prod }{\partial w_{0}}=0$$

Substituting Equations (38) and (40) into Equation (35), the total energy equation can be obtained:(47)$$\prod \left(u_{0},w_{0}\right)=\frac{D_{1}\left(z\right)}{2}\frac{\pi^{4}}{a^{2}}w_{0}{}^{2}+\frac{C_{1}\left(z\right)}{2}\left[\left(1-\upsilon {\left(z\right)}^{2}\right)u_{0}{}^{2}-\frac{2u_{0}w_{0}^{2}\left(1-\upsilon {\left(z\right)}^{2}\right)}{8a}+\frac{\left(3-2v\right)\pi^{4}w_{0}{}^{4}}{64a^{2}}\right]-Pu_{0}$$

Therefore Equations (45) and (46) can be reduced to
(48)$$P=\left(1-\upsilon {\left(z\right)}^{2}\right)C_{1}\left(z\right)\left(u_{0}-\frac{\pi^{2}}{8}\frac{w_{0}{}^{2}}{a}\right)$$
(49)$$64{\left(\frac{\pi }{a}\right)}^{2}w_{0}\left[\frac{4\pi^{2}D_{1}\left(z\right)}{C_{1}\left(z\right)}-\left(1-\upsilon {\left(z\right)}^{2}\right)au_{0}+\left(3-2\upsilon \left(z\right)\right)\frac{\pi^{2}}{8}w_{0}{}^{2}\right]=0$$

According to Equation (49), the relationship between out-plane and in-plane displacement can be written as:(50)$$w_{0}{}^{2}=\frac{8u_{0}a\left(1-\upsilon {\left(z\right)}^{2}\right)}{\pi^{2}\left(3-2\upsilon \left(z\right)\right)}-\frac{32D_{1}\left(z\right)}{C_{1}\left(z\right)\left(3-2\upsilon \left(z\right)\right)}$$

Considering the pre-buckling problem, when $w_{0}=0$, Equations (48) and (50) can be derived to
(51)$$P=\left(1-\upsilon {\left(z\right)}^{2}\right)C_{1}\left(z\right)u_{0}=E_{1}\left(z\right)hu_{0}$$
(52)$${\left(u_{0}\right)}_{c}=\frac{\pi^{2}h^{2}}{3a\left(1-\upsilon {\left(z\right)}^{2}\right)}$$

From Equation (52), it can be seen that the scale effect parameter does not affect the critical buckling displacement.

Eliminating $w_{0}$ based on Equations (48) and (49), we can obtain an approximate linear post-buckling solution as follows:(53)$$P=\frac{13}{25}\left(1-\nu^{2}\right)C_{1}u_{0}+\frac{\left(1-\nu^{2}\right)4\pi^{2}D_{1}}{\left(3-2\nu \right)a}$$
where ${\left(u_{0}\right)}_{c}$ is the critical displacement in the *x*-direction. The critical buckling force $P_{c}{}_{r}$ can be obtained by substituting Equation (52) with Equation (51).
(54)$$P_{c}{}_{r}=\frac{4\pi^{2}D_{1}\left(z\right)}{a}$$

A more specific expression is as follows:(55)$$P_{c}{}_{r}=\frac{4\pi^{2}}{a}\left\{\frac{\left((E_{U}-E_{L}){\left(\frac{z}{h}\right)}^{k}+E_{L}\right)h^{3}+6h{\left((l_{U}-l_{L}){\left(\frac{z}{h}\right)}^{k}+l_{L}\right)}^{2}\left[1-\left((\upsilon_{U}-\upsilon_{L}){\left(\frac{z}{h}\right)}^{k}+\upsilon_{L}\right)\right]}{12\left[1-{\left((\upsilon_{U}-\upsilon_{L}){\left(\frac{z}{h}\right)}^{k}+\upsilon_{L}\right)}^{2}\right]}\right\}$$

## 4. Numerical Results

This section investigates numerical examples of pre-buckling and post-buckling problems for FGM thin plates. The FGM is made of Al and Cu. The upper surface is Cu and the lower surface is Al. The material properties of the FGM are described as shown in Table 1.

### 4.1. Effects of the Scale Parameter

By considering various scale parameters, the relationship between force and x-direction displacement is plotted in Figure 3 based on CT and MCST. Equation (23) shows that the MCST can be converted into CT when scaling parameter *l* = 0. It can be seen from Figure 3 that when $u_{0}<u_{0}{}_{c}$, it represents the thin plate in the pre-buckling state. Conversely, $u_{0}>u_{0}{}_{c}$ represents the post-buckling state. Figure 3 shows that with the same x-direction displacement, as the scale parameter *l* increases, the force P tends to increase. The increase in image slope with the increase in scale effect parameter indicates the increase in flexural stiffness of the thin plate. At the instant of buckling, the buckling load *P* decreases. The slope of the post-buckling stage is smaller than that of the pre-buckling stage, indicating that the flexural stiffness of the thin plate decreases after buckling. According to Equation (53), consider a 10 μm thick Cu plate and Al plate. The Cu plate flexural stiffness post-buckling is 0.52 times that pre-buckling based on CT, and 0.5204 times based on MCST. The Al plate flexural stiffness post-buckling is 0.52 times that pre-buckling based on CT, and 0.53 times based on MCST. The details are shown in Table 2, where D_pre_ is the pre-buckling flexural stiffness and D_post_ is post-buckling flexural stiffness.

For the critical buckling displacement results, the linear analysis in Figure 3 and the nonlinear simulation in Figure 4 are approximately the same. However, the critical load for nonlinear buckling is significantly smaller than that for linear buckling. Linear buckling obtains the critical buckling load based on the small deformation assumption. In the nonlinear buckling analysis, when the load is small, the analysis results are consistent with the linear analysis. As the load gradually increases, the deformation increases, the structure exhibits nonlinear response, and the force-displacement curve begins to deviate from the linear result. Therefore, the critical buckling load is less than the linear analysis result.

The simulation results of buckling are shown in Figure 5 and Figure 6. The parameters of the model are a micro-copper plate with length 200 μm, width 200 μm, thickness 10 μm, Young’s modulus 110 Gpa, yield strength 230 Mpa and Poisson’s ratio 0.35.

Figure 5 shows the linear buckling displacement response of Cu plates. The whole deformation is regular, and the maximum out-of-plane displacement is in the center of the plate.

Figure 6 shows the irregular stress distribution in post-buckling nonlinear analysis. It also exhibits irregular deformation compared with the linear post-buckling analysis. There is maximum stress at the center of the Cu plate in the nonlinear post-buckling analysis, which is also the most prone to failure. The minimum stress is in the middle of the edge of the Cu plate.

Figure 7 shows that with the same deflection, as the scale parameter *l* increases, the buckling strain energy tends to increase. The buckling strain energy increases as the deflection increases. Figure 8 shows the two possibilities for buckling: in-plane, and out-of-plane buckling. Figure 9 is the contour line of membrane energy. Figure 8 and Figure 9 indicate the variation of membrane energy with x-direction displacement and deflection.

### 4.2. Effects of Power-Law Index Parameter k

By considering various power-law index parameters k, the changes in Young’s modulus, Poisson’s ratio, and density through the thickness direction of the FGM plate are investigated and shown in Figure 10, Figure 11 and Figure 12 for different material distributions. The power-law index parameter k has an important effect on the Young’s modulus and Poisson’s ratio of the FGM thin plate. Figure 13 shows that with the same material proportions, the critical displacement in the *x*-direction decreases with the decreasing power exponent *k*. At the same power exponent *k*, the critical displacement in the *x*-direction decreases as the proportion of the upper material increases. According to Equation (52), the scale effect does not change the critical displacement in the *x*-direction. However, power-law index parameter *k* has a significant effect on it.

### 4.3. Influence of Power-Law Distribution Considering the Scale Effect

According to Equations (39) and (50), the relationship of buckling strain energy, *x*-direction displacement and material proportion can be seen in Figure 14. The power-law index parameters have a significant effect on buckling strain energy. When the material ratio is determined, the value of the scale effect parameter is determined, and the buckling energy is determined accordingly. Figure 14 shows that the buckling strain energy increases with the increase in *x*-direction displacement. At the same proportion of material and *x*-direction displacement, the buckling strain energy increases with the increasing power-law index *k.* The membrane energy decreases as the power-law index parameter increases, as is shown in Figure 15.

The significance of the power-law index parameters *k* and scale effect parameters *l* is shown in Figure 16, Figure 17 and Figure 18. However, compared with power-law index parameters *k*, the scale effect parameters have a greater influence on critical buckling load. From Figure 16, with the same scale parameter *l* and power exponent *k*, the critical buckling load varies little with the material ratio. Figure 15 shows that the higher the modulus of elasticity, the higher the critical buckling load. When *l* = 0, the maximum difference of critical buckling load is less than 0.5 N. However, it shows a large difference in Figure 18 when considering the power-law distribution of the scale effect. The maximum difference of the critical buckling load is more than 30 N, as shown in Figure 18. The scale effect parameter of the lower layer of Al is about 4.6 times that of the upper layer of Cu. Therefore, as the proportion of the lower layer Al material increases, the scale effect parameter of the FGM plate also increases, thus leading to a significant increase in the critical buckling load.

Table 3 shows the critical buckling loads P_cr_ of FGM plates under different theories. FGM plates are made of metal (SUS304) and ceramic (Si_3_N_4_). The properties of FGM have been determined as follows [59]:E(SUS304) = 201.04 GPa, *υ*(SUS304) = 0.3262, *ρ*(SUS304) = 8166 kg/m^3^
E(Si_3_N_4_) = 348.43 GPa, *υ*(Si_3_N_4_) = 0.2400, *ρ*(Si_3_N_4_) = 2370 kg/m^3^

Present I represents the critical buckling load without considering the scale effect. The critical buckling load decreases as the power law index parameter increases. With the increase in the power law index, the volume fraction of SUS304 increases. The increase in the volume fraction of SUS304 results in a decrease in the hardness of the FGM plates, resulting in a decrease in the critical buckling load. Present II and Present III represent the critical buckling loads when considering a power-law distribution with scale effects. There are no experimental data and literature on the scale effect parameters of FGM, so we assume a scale effect parameter of 1 μm for SUS304 and a scale effect parameter of 2 μm for Si_3_N_4_ in Present II, and assume a scale effect parameter of 2 μm for SUS304 and a scale effect parameter of 1 μm for Si_3_N_4_ in Present III as a comparison. Present II shows that when the Si_3_N_4_ volume fraction increases, the elastic modulus and scale effect parameters of the FGM increase, resulting in an increase in the critical buckling load compared with Present I. Present III shows that when the Si3N4 volume fraction increases, the elastic modulus of the FGM increases but the scale effect parameter decreases, resulting in less fluctuation in the critical buckling load.

## 5. Concluding Remarks

Based on the modified coupled stress theory, the post-buckling of an Al-Cu FGM thin plate is analyzed. According to Hamilton’s principle, deflection solutions and stress solutions for post-buckling under-scale effects are derived. The influence of the scale effect on the buckling strain energy and membrane energy is investigated in detail. The most significant conclusions can be summarized as follows:Under the same conditions, the critical buckling load, the membrane energy, and buckling strain energy increase as the scale effect parameter increases. The critical *x*-direction displacement is not affected by scale effects.The critical buckling displacement for nonlinear buckling is consistent with the linear buckling analysis, but the critical buckling load is smaller considering the nonlinearity of the structureUnder the same circumstances, the modulus of elasticity, Poisson’s ratio, critical buckling load, critical buckling displacement, and buckling strain energy all increase with increasing power-law index parameter *k*. The membrane energy decreases as the power-law index parameter increases. As the power-law index *k* increases, the volume fraction of bottom copper increases. The increased modulus of elasticity of the FGM plate results in an increase in stiffness and thus an increase in the critical buckling load.Both the scale effect parameter *l* and the power-law exponent *k* have important effects on the FGM thin plate buckling and post-buckling problems, but in comparison, the scale effect parameter is more influential than the power-law exponent.The scale effect parameter can greatly increase the critical buckling load, and the correct choice of material is important for the stable design of the device. The scale effect cannot be ignored, and the research on the scale effect is of great significance for MEMS/NEMS design, and manufacture.

It is of great significance to consider continuously distributed scale effect parameters for micro-scale FGM. Both scale effect parameters and power-law index parameters have important effects on the mechanical behavior of micro-scale FGM. The scale effect parameter affects the stiffness of the material, and the power-law index parameter affects the intrinsic properties of the FGM. The present results will be used for the design of micro-scale FGM MEMS/NEMS, and it is necessary to consider both the MSCT and Power-law in the design of micro-scale FGM.

## Figures and Tables

**Figure 1 nanomaterials-12-03502-f001:**
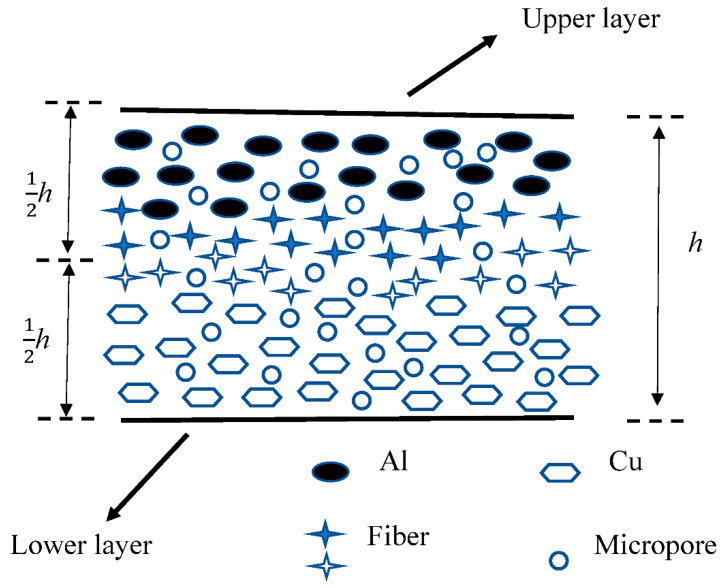
Schematic diagram of a cross-section of Al–Cu functionally graded material distributed along the thickness direction.

**Figure 2 nanomaterials-12-03502-f002:**
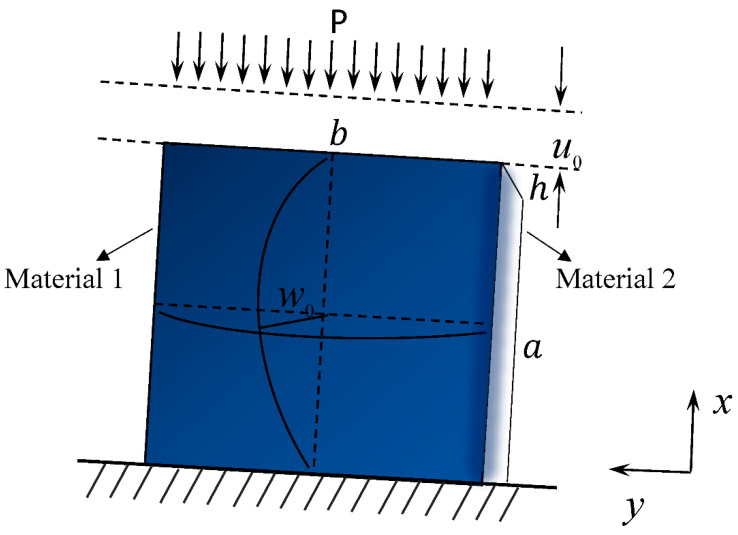
Loading and geometry of an FGM thin rectangular plate model.

**Figure 3 nanomaterials-12-03502-f003:**
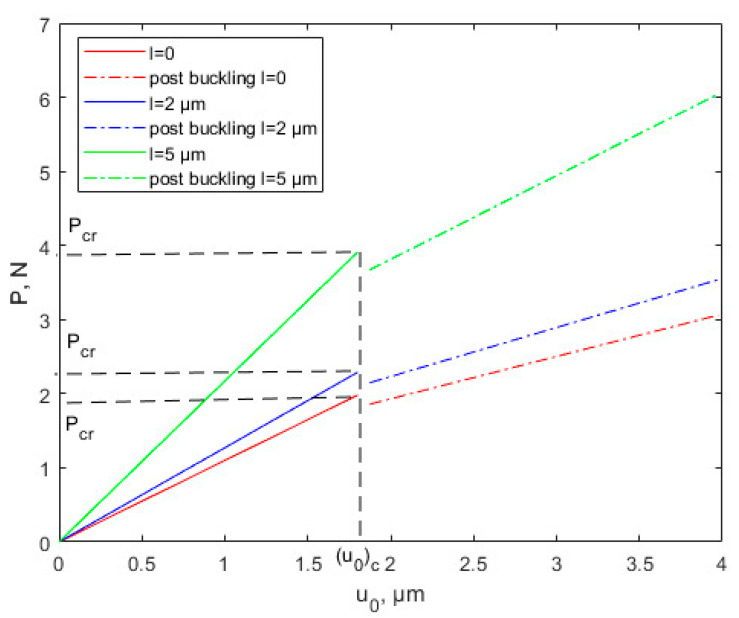
Pre-buckling and linear post-buckling responses of thin Cu plate based on CT and MCST.

**Figure 4 nanomaterials-12-03502-f004:**
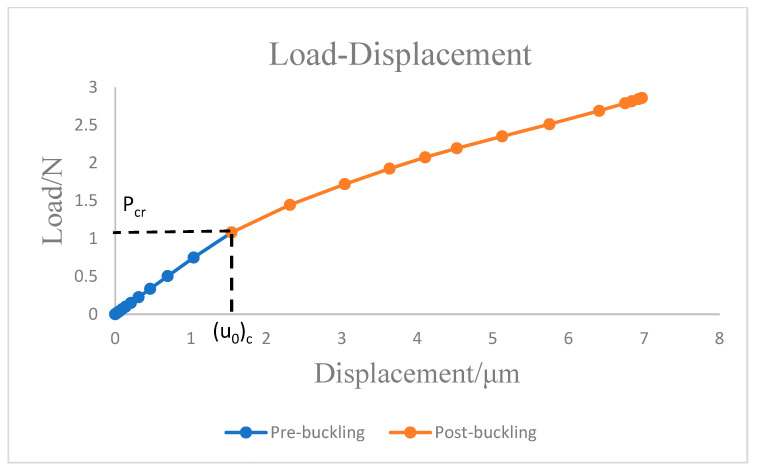
Nonlinear buckling response simulation of Cu plates.

**Figure 5 nanomaterials-12-03502-f005:**
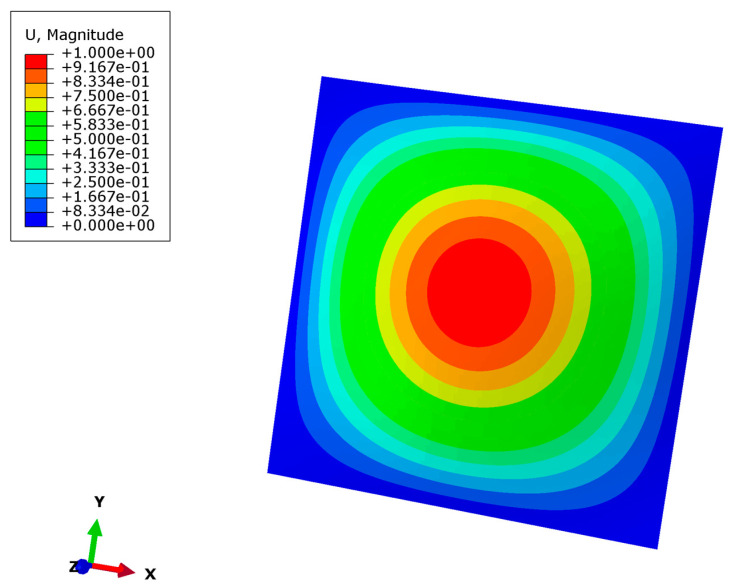
Linear buckling displacement response simulation of Cu plates.

**Figure 6 nanomaterials-12-03502-f006:**
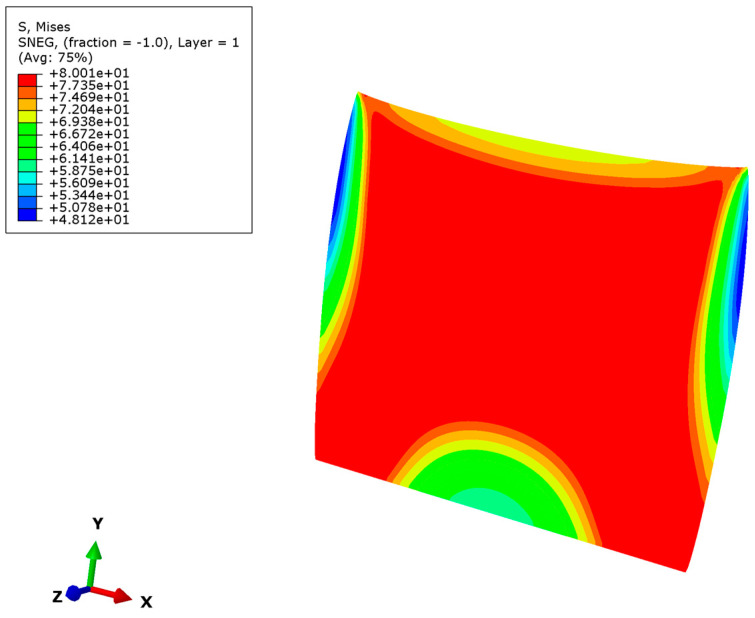
Nonlinear buckling stress response simulation of Cu plates.

**Figure 7 nanomaterials-12-03502-f007:**
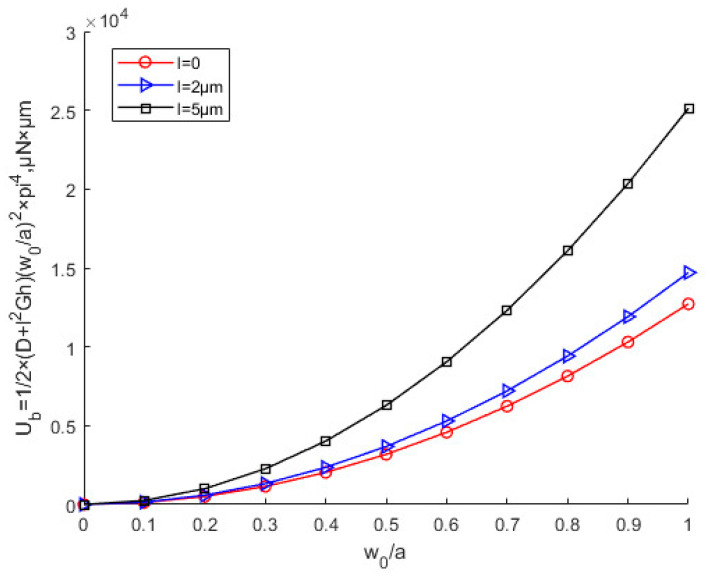
Effects of the scale parameters on buckling strain energy of a thin Cu plate.

**Figure 8 nanomaterials-12-03502-f008:**
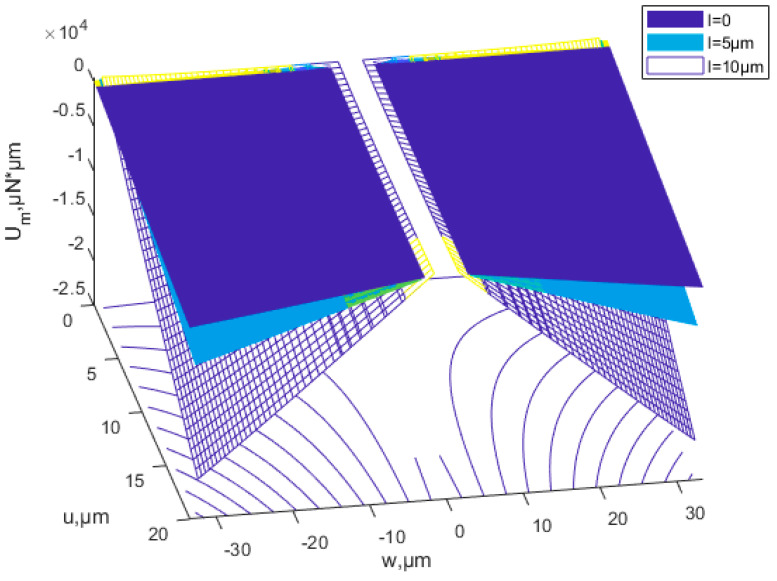
Effects of the scale parameters on membrane energy of a thin Cu plate.

**Figure 9 nanomaterials-12-03502-f009:**
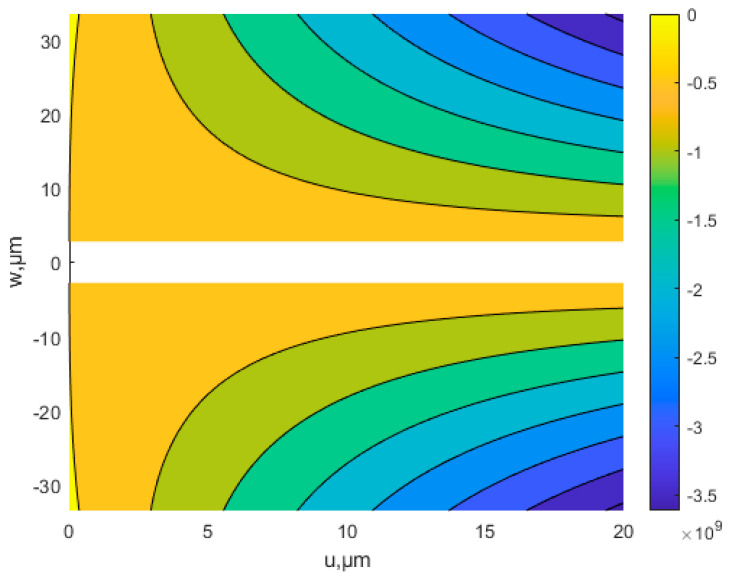
Contour line of membrane energy.

**Figure 10 nanomaterials-12-03502-f010:**
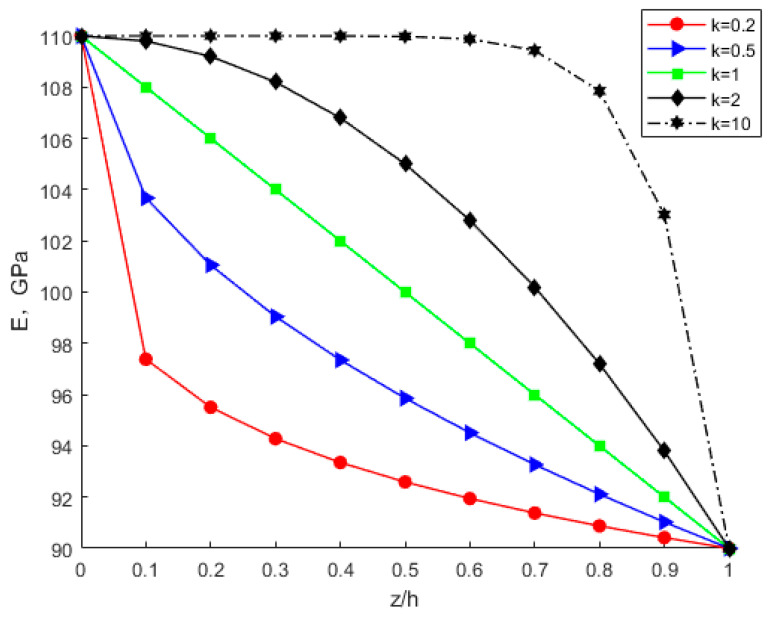
Variation in Young’s modulus through the thickness direction of the FGM plate.

**Figure 11 nanomaterials-12-03502-f011:**
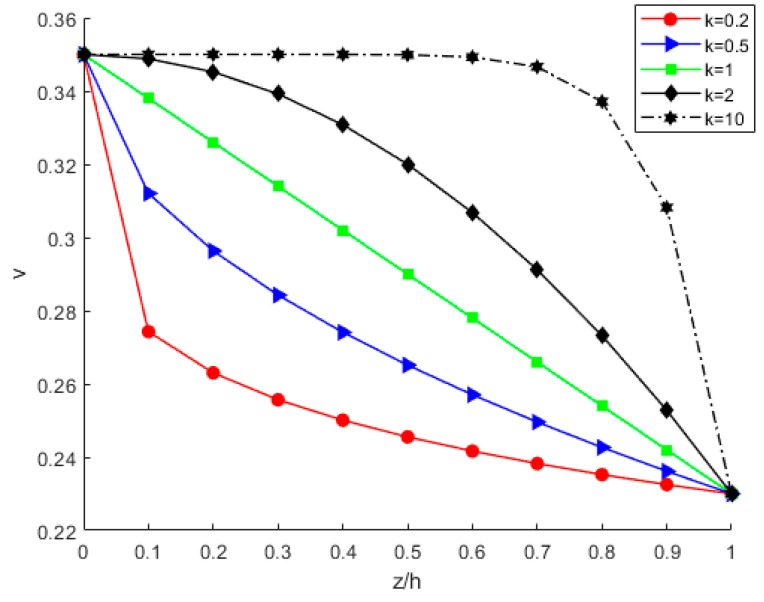
Variation in Poisson ratio through the thickness direction of the FGM plate.

**Figure 12 nanomaterials-12-03502-f012:**
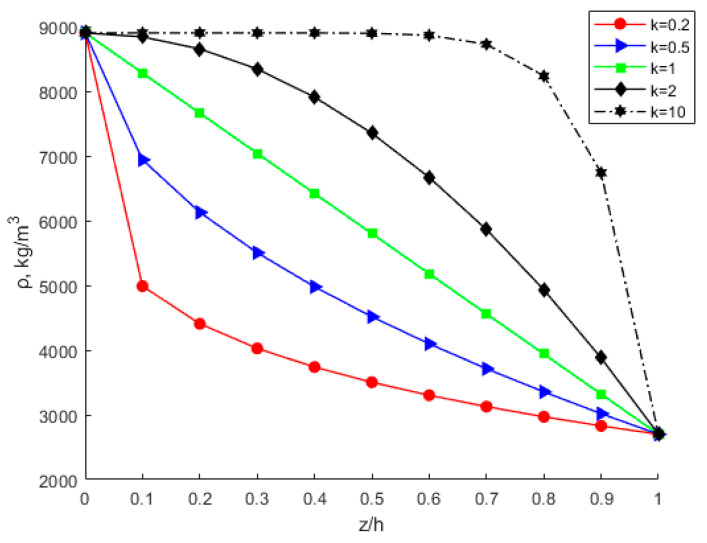
Variation in density through the thickness direction of the FGM plate.

**Figure 13 nanomaterials-12-03502-f013:**
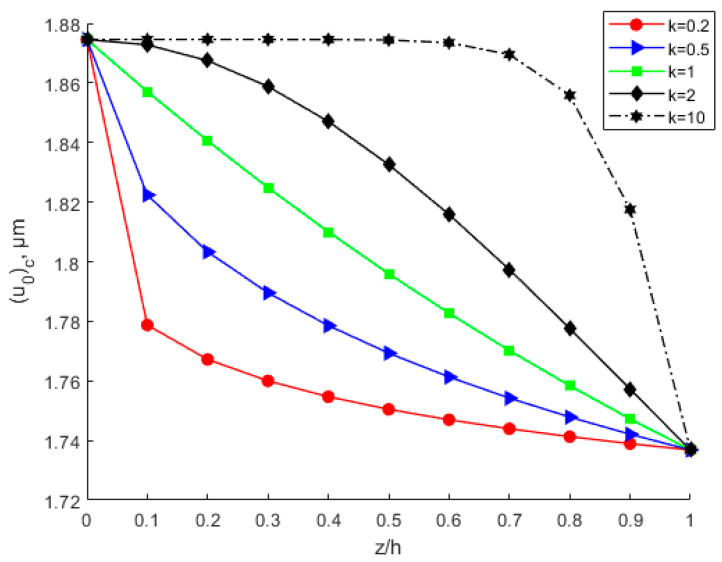
Variation in the x-direction critical displacement through the thickness direction of the FGM plate.

**Figure 14 nanomaterials-12-03502-f014:**
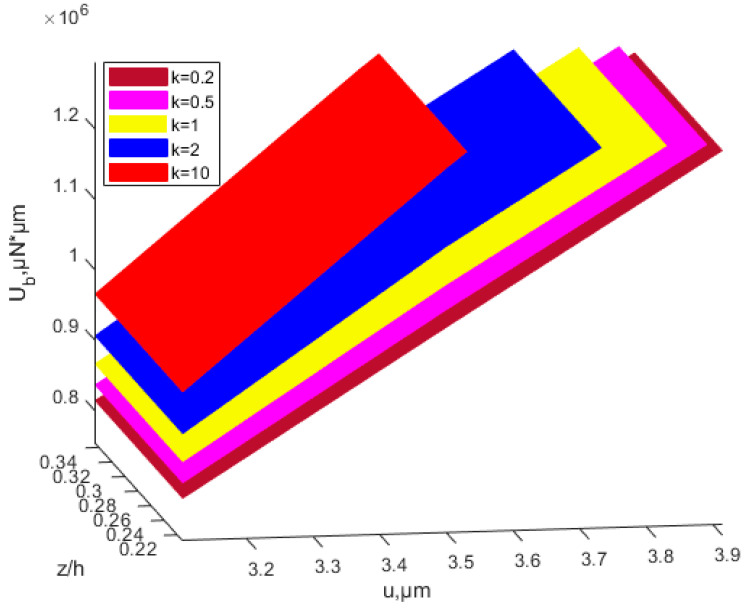
Influence of the power-law distribution of the scale effect on buckling strain energy of the FGM plate.

**Figure 15 nanomaterials-12-03502-f015:**
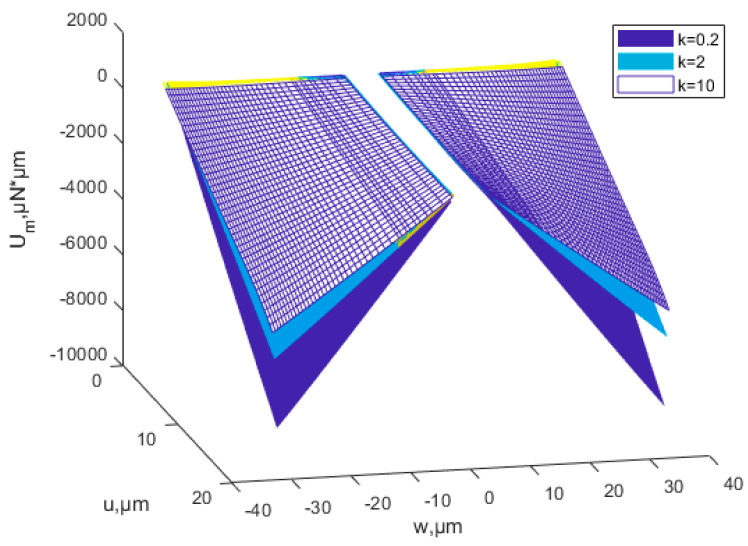
Influence of the power-law distribution of the scale effect on membrane energy of the FGM plate.

**Figure 16 nanomaterials-12-03502-f016:**
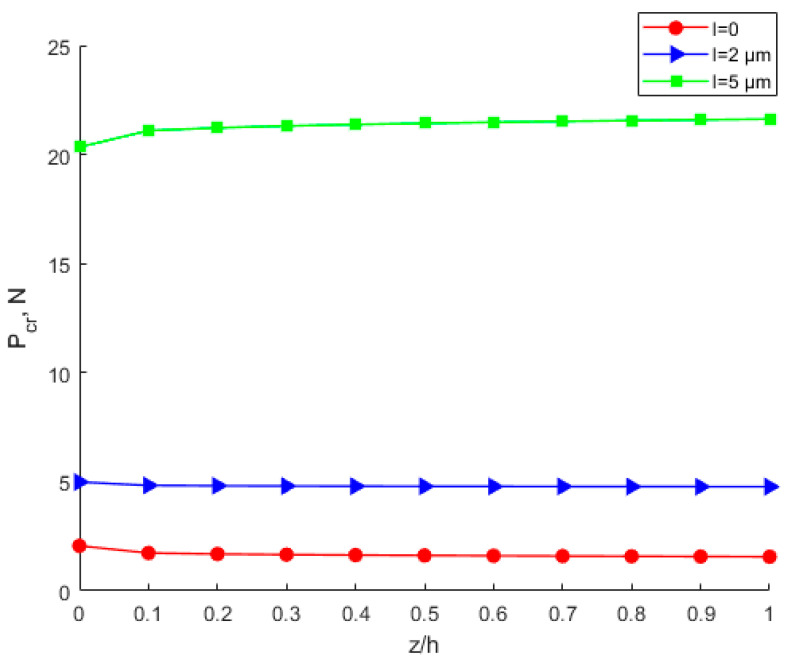
Influence of scale effect parameters on the critical buckling load when *k* = 0.2.

**Figure 17 nanomaterials-12-03502-f017:**
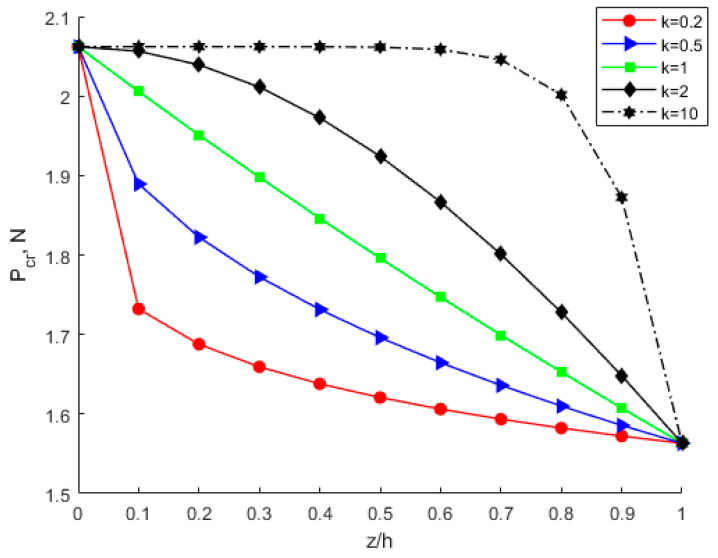
Influence of power-law index parameters on the critical buckling load when *l* = 0.

**Figure 18 nanomaterials-12-03502-f018:**
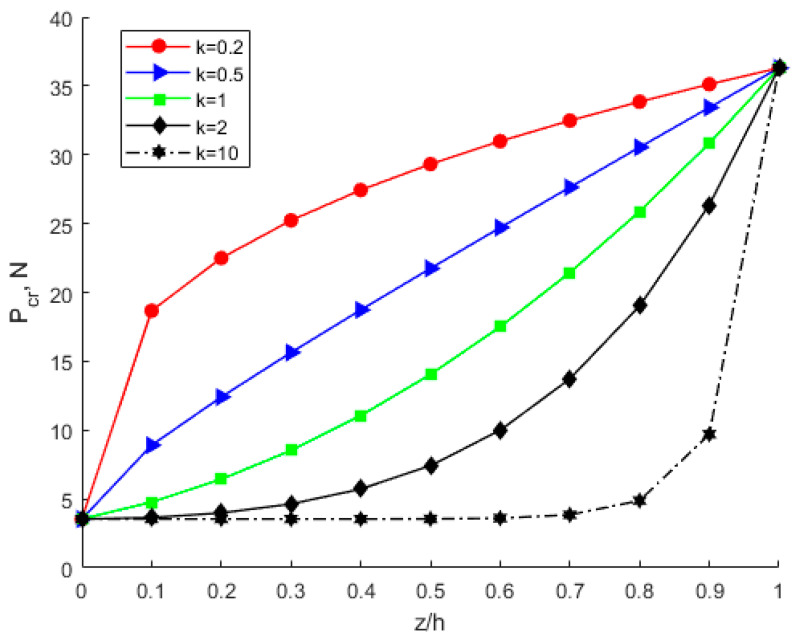
Influence of power-law distribution of scale effect on the critical buckling load.

**Table 1 nanomaterials-12-03502-t001:** Material properties of Cu and Al.

Materials	*E*	*υ*	*ρ*	*l* [16]	*h*	*a*	*b*
Cu	110 GPa	0.35	8900 kg/m^3^	1.422 μm	10 μm	200 μm	200 μm
Al [32]	90 GPa	0.23	2700 kg/m^3^	6.58 μm	10 μm	200 μm	200 μm

**Table 2 nanomaterials-12-03502-t002:** Cu and Al plate flexural stiffness between pre-buckling and post-buckling based on CT and MCST.

Materials	D_pre_ of CT	D_post_ of CT	D_pre_ of MCST (*l* = 2 μm)	D_post_ of MCST (*l* = 2 μm)	D_pre_ of MCST (*l* = 5 μm)	D_post_ of MCST (*l* = 5 μm)
**Cu**	110 × 10^4^ N/m	57.2 × 10^4^ N/m	127.16 × 10^4^ N/m	66.12 × 10^4^ N/m	217.25 × 10^4^ N/m	112.86 × 10^4^ N/m
**Al**	90 × 10^4^ N/m	46.8 × 10^4^ N/m	104.04 × 10^4^ N/m	54.10 × 10^4^ N/m	177.75 × 10^4^ N/m	92.43 × 10^4^ N/m

**Table 3 nanomaterials-12-03502-t003:** Critical buckling load Pcr for the FGM plates subjected to uniaxial compressive loads in the *x*-direction.

Theory	*a*/*b*	*a/h*	*k*				
			0	0.5	1	5	10
ESDPT [60,61]	1	20	8.1510	6.9310	6.4710	5.8410	5.6010
3DT [59]	1	20	8.1099	6.8699	6.3899	5.7299	5.4899
Bateni M et al. [62]	1	20	9.6507	8.3142	7.8183	7.1039	-
Present I	1	20	6.0820	5.2508	4.9043	4.1837	4.0245
Present II	1	20	9.2650	6.6177	5.9370	5.3573	5.0588
Present III	1	20	6.8780	6.7276	6.6973	6.6842	6.6882

## Data Availability

Not applicable.

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
