# Peer review of "Size-Dependent Buckling and Post-Buckling Analysis of the Functionally Graded Thin Plate Al–Cu Material Based on a Modified Couple Stress Theory"

_nanomaterials, 2022, doi:10.3390/nano12193502_

Round 1

Reviewer 1 Report

The manuscript is prepared good and worth considering for publication. The authors are required to address the following comments:

1. What is the influence of interface between two Al-Cu plate materials and its defects on the FGM properties?

2. The paper is only based on numerical study. How accurate is the obtained results for designing the FGM materials in experiment? This comment needs to be addressed in a satisfactory way. 

3. What is the influence of porosity on the mechanical properties of the proposed FGM structure? "The authors are suggested to use and cite some of the published papers on Functionally graded metal syntactic foams for this purpose".

4. The schematic of the FGM structure in the current study must be included in the beginning of the paper.

Reviewer 2 Report

This paper investigated size-dependent buckling and postbuckling of functionally graded microplates using a modified couple stress theory. The topic has been studied widely, thus the significance of this paper remains to be discussed. The following grammatical mistakes and technical issues need to be fully corrected and addressed before the reconsideration.

1.              Abstract: Grammar error: “Comparing the numerical analysis of linear buckling with the simulation analysis of nonlinear buckling.” Please rewrite this sentence.

Grammar error: “The influence of scale effect parameters l and power-law index parameters k on buckling and post-buckling problems are analyzed.”

The authors showed many specific detailed results in the abstract, but without presenting the major conclusion.

2.              Extensive literature review was performed in Introduction, but their limitations and facing challenges were not pointed out. The motivation and novelty of this study should be highlighted.

The FGM microplate in this paper is composed of Al and Cu. Graphene is a kind of 2D material which can be used as nanofiller to reinforce metal materials to form high-performance metal matrix nanocomposites due to its very excellent mechanical properties. Accordingly, graphene-based nanocomposites can be designed to be FGMs with graphene content being varying in a graded pattern along one direction. With the introduction of graphene in FGM composite structures, their structural behaviors including buckling, postbuckling, free vibration, dynamic instability, etc., can improve significantly. Furthermore, incorporating graphene origami into FGMs can also lead to unique mechanical properties such as negative Poisson’s ratio, giving rise to superior structural performances and multifunctionalities, which will have huge application potentials in MEMS/NEMS devices. Therefore, more discussions on the functionally graded graphene reinforced composite structures and functionally graded graphene origami-enabled auxetic metamaterial structures are advised to be carried out in Introduction.

3.              Young's modulus and Poisson's ratio are changed in a gradient pattern along the plate thickness, how about density?

4.              The numerical results should be validated first by comparing the present results with other published results before performing the parametric study. Otherwise, how do you make sure the validity and correctness of the results presented in the paper?

5.              Page 11, format is not the same for these two paragraphs.

Figs 4 and 5, the displacement and stress distribution of plate are calculated based on Abaqus software. But the FEM model details are not given. The image quality of these two figures should be improved. There is Chinese character in Fig 5 that should be avoided in English journal.

The authors only described the results presented in figures, but did not show the in-depth understanding on the reason behind the phenomenon.

Reviewer 3 Report

From my point of view, the principal insufficient of this research is thethesis (see 2. Formulations and theories of FGM with MCST 2.1. Power-lower of FGM)

"The gradient change of material properties can be expressed as a power-law function as: //formula (1)//"

I do not know and did not find arguments to the favor of this thesis.

Now, it looks like an authors' assumption. etc.

It is necessary to say clear about the status of the law given by formula (1). If it is a fundamental and justified material property, it must be supported by the proper explanation and the references.

If the formula (1) is the authors' assumption, I am afraid, the paper has no/very low level of scientific interest.

If the change of material properties can be expressed as a power-law function, the paper has a scientific value and may be estimated after the proper explanation/information will be added.

If the paper will be resubmitted, I strongly recommend include a figure, demonstrating the cross-section of the plate with indication of the thickness of the Cu and Al layers and interface zone.

Round 2

Reviewer 2 Report

This updated version has addressed my raised concerns.

Reviewer 3 Report

My comments were accounted. So, I accept the paper